# The AAA-ATPase Ter94 regulates wing size in *Drosophila* by suppressing the Hippo pathway
Mingming Li[1,3], Wenhao Ding[1,3], Yanran Deng[2], Yunhe Zhao[1], Qingxin Liu ⓘ [1] ✉ & Zizhang Zhou ⓘ [1,2] ✉

Insect wing development is a fascinating and intricate process that involves the regulation of wing size through cell proliferation and apoptosis. In this study, we find that Ter94, an AAA-ATPase, is essential for proper wing size dependently on its ATPase activity. Loss of Ter94 enables the suppression of Hippo target genes. When Ter94 is depleted, it results in reduced wing size and increased apoptosis, which can be rescued by inhibiting the Hippo pathway. Biochemical experiments reveal that Ter94 reciprocally binds to Mer, a critical upstream component of the Hippo pathway, and disrupts its interaction with Ex and Kib. This disruption prevents the formation of the Ex-Mer-Kib complex, ultimately leading to the inactivation of the Hippo pathway and promoting proper wing development. Finally, we show that hVCP, the human homolog of Ter94, is able to substitute for Ter94 in modulating *Drosophila* wing size, underscoring their functional conservation. In conclusion, Ter94 plays a positive role in regulating wing size by interfering with the Ex-Mer-Kib complex, which results in the suppression of the Hippo pathway.

Wings are vital organs for insects, which participate in multiple processes, such as foraging and mating. Unlike the wings of birds, insect wings are usually formed through metamorphosis from the wing discs in larva[1,2]. This process involves cell proliferation and apoptosis, which are controlled by the Hippo pathway[3,4]. The core of the Hippo pathway consists of a series of kinases and the transcriptional cofactor Yorkie (Yki)[5,6]. The Ste20-like protein kinase Hippo (Hpo) complexes with Salvador (Sav) to phosphorylate and activate the downstream kinase Warts (Wts). Next, with assistance of the adaptor Mats, Warts directly binds and phosphorylates Yki on several serine residues, leading to its cytoplasmic retention[7,8]. When this kinase cascade is inactivated, unphosphorylated Yki enters the nucleus to turn on the expression of target genes[9]. As a matter of fact, due to the lack of DNA-binding domains, Yki only works synergistically with other transcription factors, most notably Sd[10,11]. In general, Yki enables the promotion of cell division and the suppression of cell death by activating the expression of pro-proliferative and anti-apoptotic genes[12,13]. In addition to inhibiting Yki's entry into the nucleus, our recent studies also reveal that the Hippo pathway promotes Yki degradation through dissociating the Yki-Usp7 interaction[14]. Thus, the Hippo pathway governs Yki's activity, at least through dual mechanisms, to regulate cell proliferation and death. In *Drosophila*, abnormal

expression of any component of the core kinase cassette leads to wing defects, reflecting its important role for wing development[15,16].

In *Drosophila*, there are two main branches upstream of the core kinase cassette: the Fat (Ft)-Dachsous (Ds) complex and the Ex (Expanded)-Mer (Merlin)-Kib (Kibra) complex[5]. Both Ft and Ds are transmembrane cadherins, that can independently activate the Hippo pathway, or work together to strengthen signaling through cell-cell contacts[17]. Ex, Mer and Kib form a ternary complex localized to the apical domain of epithelial cells to activate the Hpo-Wts kinase cascade[18,19]. Loss of Mer and Kib alone in *Drosophila* eyes fails to cause obvious defects, while their simultaneous loss leads to severe overgrowth, accompanied by activation of Hippo-responsive genes[18]. Another study has demonstrated that Mer and Ex work synergistically to activate the Hippo pathway[20], emphasizing the importance of the Ex-Mer-Kib complex. Among all components of mammalian Hippo pathway, only NF2, the ortholog of Mer, undergoes heavy ubiquitin modification[21]. BRCA1/BARD1-mediated ubiquitination of NF2 does not promote its degradation, but rather influences its interactions with partners[21]. Moreover, NEDD4L-mediated Merlin ubiquitination on K396 promotes its binding to the downstream kinase Lats1, leading to the activation of the Hippo pathway[22]. Thus, ubiquitin modification on Mer always plays non-degradative roles. Due to the

[1]College of Life Sciences, Shandong Agricultural University, Tai'an, China. [2]Key Laboratory of Biodiversity Conservation and Bioresource Utilization of Jiangxi Province, College of Life Sciences, Jiangxi Normal University, Nanchang, China. [3]These authors contributed equally: Mingming Li, Wenhao Ding. ✉ e-mail: liuqingxin@sdau.edu.cn; zhouzz@sdau.edu.cn

importance of Mer ubiquitination in the activation of the Hippo pathway, the mechanism regulating the activity of ubiquitin-modified Mer remains unclear but is crucial.

Ter94, also known as p97 or valosin-containing protein (VCP), is an evolutionarily conserved chaperone-like AAA+ ATPase that is widely expressed in eukaryotic cells[23]. It was initially identified for its involvement in endoplasmic reticulum-associated degradation[24]. Recent studies have gradually uncovered that Ter94 plays multiple roles in various cellular processes. One of its functions is the remodeling of chromatin through its ATPase activity, thereby facilitating DNA transcription[25,26]. Ter94 also participates in modulating RNA splicing and polyadenylation to control RNA metabolism[27]. Generally, Ter94 recognizes ubiquitin-modified proteins and extracts them for further processing[28,29]. The ATPase activity of Ter94 is dispensable for this extraction, as it requires energy to generate mechanical force[30,31]. In the cytoplasm, Ter94 specifically recognizes K11-linked ubiquitinated Ci, guiding it to proteasomes for partial degradation[32]. When cells are exposed to ultraviolet light, Ter94 facilitates the degradation of ubiquitin-modified XPC, triggering DNA damage[33]. In addition, Ter94 extracts K6-linked ubiquitinated c-MYC from the c-MYC-MAX heterodimer for subsequent proteasomal degradation[34]. Besides, Ter94 disassembles the PP1-SSD22-I3 inhibitory complex to activate PP1's phosphatase activity, without affecting their protein levels[35,36]. Thus, Ter94 plays both degradative and non-degradative roles for ubiquitin-modified proteins.

In this study, we conducted an RNA interference (RNAi) screening in *Drosophila*, and identified that knockdown of Ter94 decreased wing size. Loss of Ter94 inhibited the expression of Hippo-responsive genes and triggered apoptosis in wing discs. In addition, we demonstrated that the AAA+ ATPase activity of Ter94 is crucial for its role in regulating the Hippo pathway. Interestingly, human VCP was able to functionally replace Ter94 in controlling wing size, Yki target gene expression, and apoptosis, reflecting their conservation. Through epistatic analyses, we determined that Ter94 located upstream of the core kinase cascade in modulating the Hippo pathway. Mechanistically, Ter94 bound Mer to weaken its interaction with Ex or Kib, without affecting the abundance of Mer. Thus, the disruption of the Ex-Mer-Kib complex by Ter94 leads to the suppression of the Hippo pathway. In summary, our study reveals that Ter94 suppresses the Hippo pathway by interfering with the formation of the Ex-Mer-Kib complex, thereby modulating wing size.

## Results

### Knockdown of *ter94* decreases wing size

The *Drosophila* wing provides an ideal model for screening genes that determine organ size. To identify genes that control wing size, we crossed wing-specific *nub*-gal4 flies with RNAi lines to silence gene expression, and then observed wings of the offspring. Our unbiased screening revealed Ter94 as a potential regulator of wing size. Compared to the control RNAi (Fig. 1a), knockdown of *ter94* using two RNAi lines from TsingHua Fly

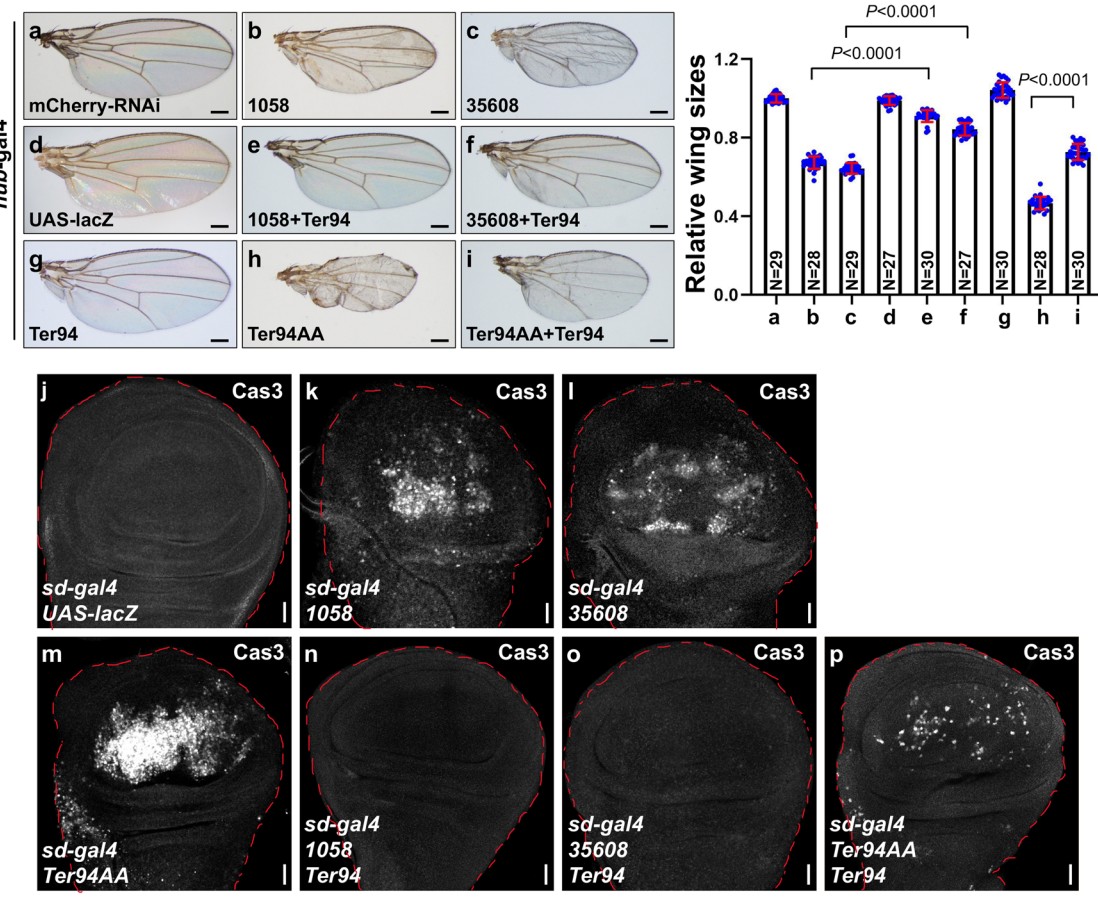

**Fig. 1 | Knockdown of *ter94* or overexpression of Ter94AA decreased wing size. a–i** Comparison of adult wings from control flies **a**, **d**, knockdown of *ter94* by *nub*-gal4 **b**, **c**, overexpressing wild-type Ter94 **g** or Ter94AA **h**, Ter94 plus *ter94* RNAi co-expression **e**, **f**, and Ter94 plus Ter94AA co-expression **i**. Quantification analyses were shown on right. The numbers in the bars represented the number of wings counted. Of note, knockdown of *ter94* or overexpression of Ter94AA apparently diminishes wing size, which is restored by expressing Ter94. **j** A control wing disc expressing UAS-lacZ via *sd*-gal4 was stained to show Cas3 (white). **k**, **l** Wing discs with *ter94* knockdown were stained to show Cas3. Notably, knockdown of *ter94* is able to activate Caspase3. **m** Overexpression of Ter94AA elevated Cas3. **n**, **p** Overexpression of Ter94 rescued the upregulation of Cas3 induced by *ter94* knockdown or Ter94AA. For all wing discs, red dotted lines mark the outlines of wing discs. Scale bars: 200 μm for all wings, 20 μm for all wing discs.

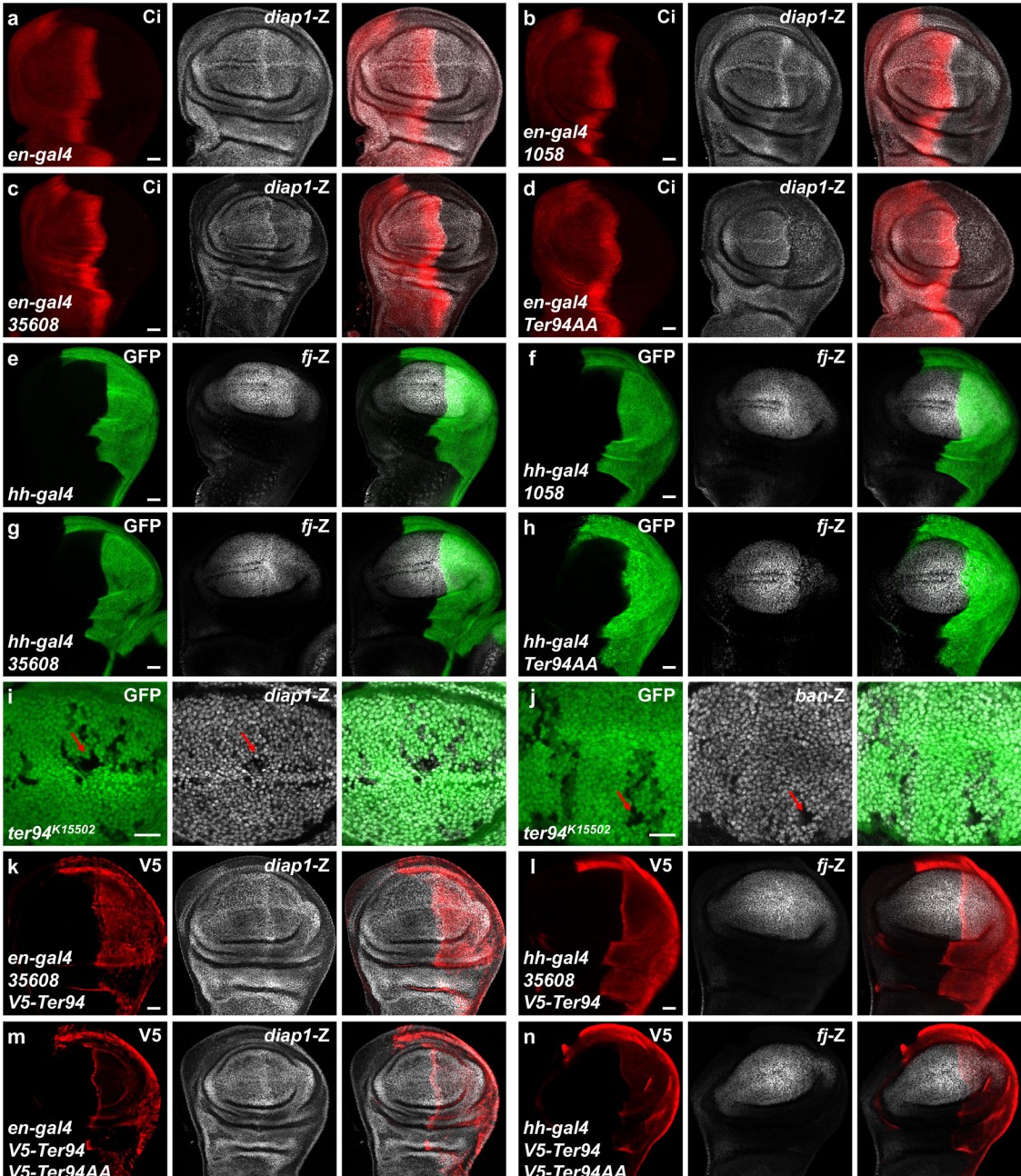

**Fig. 2 | Loss of *ter94* inhibits the expression of Yki target genes. a–d** Wing discs from control **a**, *ter94* knockdown by *en*-gal4 **b**, **c** and Ter94AA overexpression via *en*-gal4 **d** were stained with Ci (red) and *diap1*-lacZ (white). The *en*-gal4 drives UAS transgenes to express in the posterior region of the wing disc, where does not express Ci. Both knockdown of *ter94* and overexpression of Ter94AA decrease *diap1*-lacZ in wing discs. **e–h** Wing discs from control **e**, *ter94* knockdown by *hh*-gal4 **f**, **g** and Ter94AA overexpression **h** were stained to show GFP (green) and *fj*-lacZ (white). GFP marks the expression pattern of *hh*-gal4. **i-j** Wing disc carrying *ter94^{K15502}* clones were stained to show the expression of GFP (green) and *diap1*-lacZ (white in **i**) or *ban*-lacZ (white in **j**). *ter94^{K15502}* clones are recognized by the lack of GFP. Of note, *ter94* mutant cells exhibited decrease of *diap1*-lacZ (marked by arrows in **i**) and *ban*-lacZ (marked by arrows in **j**). **k, l** Wing discs simultaneously expressing V5-Ter94 and *ter94* RNAi were stained to show V5 (red) and *diap1*-lacZ (white in **k**) or *fj*-lacZ (white in **l**). Overexpression of Ter94 enables to rescue the decrease of *diap1*-lacZ and *fj*-lacZ induced by *ter94* knockdown. **m, n** Ter94 recovered the reduction of *diap1*-lacZ and *fj*-lacZ caused by Ter94AA. Scale bars: 20 μm for all wing discs.

Center (1058) and Bloomington *Drosophila* Stock Center (35608) decreased wing size (Fig. 1b-c). The reduced wing size was specifically due to decreased *ter94* expression, as introducing wild-type Ter94 was able to rescue the RNAi-induced phenotype (Fig. 1e-f). Interestingly, overexpressing wild-type Ter94 alone did not alter wing size (Fig. 1g versus Fig. 1d), suggesting that the endogenous Ter94 is sufficient to maintain proper wing size. Considering the ATPase activity of Ter94, we sought to examine whether its ATPase is involved in regulating wing size. In contrast to wild-type Ter94 (Fig. 1g), overexpression of a mutant form Ter94AA[37,38], in which the

ATP-binding sites (K248 and K521) were replaced by alanines, reduced wing size (Fig. 1h), suggesting a dominant-negative role of Ter94AA. Consistent with this view, co-expression of wild-type Ter94 to some extent rescued the small wing phenotype induced by Ter94AA (Fig. 1i). In line with this, previous studies have illustrated that Ter94AA plays a dominant-negative role in regulating the Hh[32] and Notch pathways[39], indicating that Ter94AA interferes with the endogenous Ter94 to exhibit a loss-of-function effect. To further confirm the role of Ter94 in wing size regulation, we employed the *sd*-gal4 driver to manipulate *ter94* expression in wings.

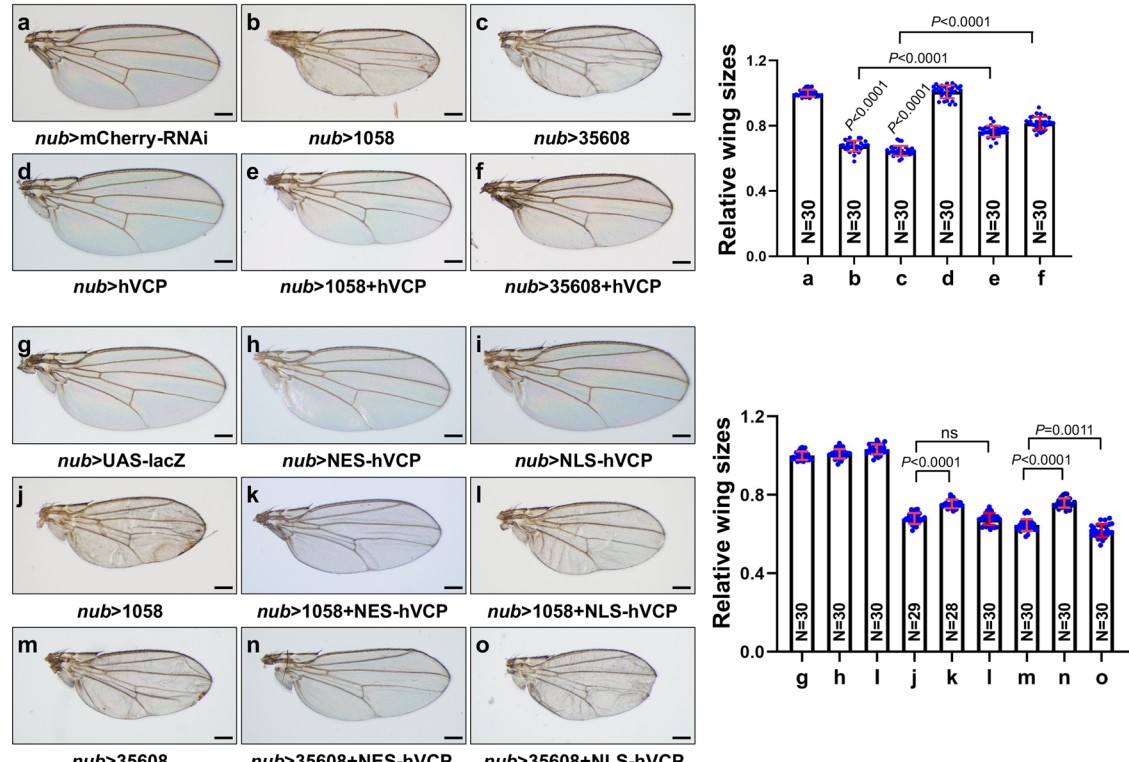

**Fig. 3 | Cytoplasmic hVCP restores Ter94-RNAi-induced small wings.**
**a**-**f** Comparison of adult wings from control flies **a**, knockdown of *ter94* by *nub*-gal4 **b**, **c**, overexpressing human VCP **d**, and simultaneous expression of human VCP plus *ter94* RNAi **e**, **f**. Quantification analyses were shown on right. The numbers in the bars represented the number of wings counted. Of note, small wings induced by *ter94* knockdown are rescued by human VCP. **g**–**o** Comparison of adult wings from control flies **g**, expressing NES-hVCP **h**, expressing NLS-hVCP **i**, *ter94* knockdown **j**, **m**, co-expression of *ter94* RNAi plus NES-hVCP **k**, **n**, and co-expressing *ter94* RNAi plus NLS-hVCP **l**, **o**. Quantification analyses were shown on right. The numbers in the bars represented the number of wings counted. Notably, small wings induced by *ter94* knockdown are rescued by NES-hVCP, not by NLS-hVCP. Scale bars: 200 μm for all adult wings.

Compared to the control wing (Supplementary Fig. 1a), both knockdown of *ter94* using 1058 (Supplementary Fig. 1b) or 35608 (Supplementary Fig. 1c) and overexpression of Ter94AA (Supplementary Fig. 1d) resulted in smaller wings. Knockdown of *ter94* using THU3262 from TsingHua Fly Center led to adult lethality, so subsequent experiments on wing size focused on using 1058 and 35608 lines. Taken together, our genetic screening identified that Ter94 positively regulates wing size in an ATPase-dependent manner.

In fact, knockdown of Ter94 not only reduced wing size, but also caused wrinkles (Fig. 1b, c), resembling cell death. To investigate this further, we used the active-Caspase3 antibody for immunostaining. Compared to the control wing disc (Fig. 1j, Supplementary Fig. 2a), both *ter94* knockdown (Fig. 1k, l, Supplementary Fig. 2b) and Ter94AA overexpression (Fig. 1m) triggered apoptosis, which could be recovered by co-expression of wild-type Ter94 (Fig. 1n, p, Supplementary Fig. 2c). On the other hand, BrdU incorporation assay showed that neither *ter94* knockdown nor Ter94AA overexpression affected cell proliferation (Supplementary Figs. 1e–h). These findings suggest that depletion of Ter94 leads to reduced wing size, at least partially through the activation of apoptosis.

### Loss of *ter94* suppresses the expression of Yki target genes
Since the Hippo pathway plays an important role in regulating organ sizes, we tried to explore whether Ter94 is involved in this pathway. In general, the Hippo pathway exerts the pro-apoptotic effect via inhibiting the activity of its transcriptional co-factor Yki[40]. Thus, we utilized several well-characterized Yki readouts (*diap1*-lacZ, *fj*-lacZ and *ban*-lacZ) to evaluate the Hippo pathway activity. Compared to the control wing disc (Fig. 2a), knockdown of *ter94* using 1058 or 35608 decreased *diap1*-lacZ levels (Fig. 2b, c). Additionally, overexpression of Ter94AA also downregulated *diap1*-lacZ (Fig. 2d), further supporting its dominant-negative effect. Similar results were obtained using another Yki readout, *fj*-lacZ (Fig. 2e–h).

Since knockdown of *ter94* using THU3262 driven by *hh*-gal4 causes lethality, so we used a temperature-sensitive ubiquitously expressed driver tub-Gal80ts to observe the change in *fj*-lacZ. As anticipated, knockdown of *ter94* using THU3262 also reduced *fj*-lacZ level (Supplementary Fig. 2d). RT-qPCR analyses showed that all of these RNAi lines effectively silenced endogenous *ter94* (Supplementary Fig. 2e). To validate these findings, we utilized a strong hypomorphic allele, *ter94*[k15502], which contains a P-element insertion disrupting Ter94 expression[32]. Homozygosity for *ter94*[k15502] was embryonic lethal, so we generated *ter94*[k15502] homozygous clones in wing discs using the Flp recombinase/Flp recombinase target (FLP/FRT) method. Analysis of these clones, marked by the loss of green fluorescent protein (GFP) signals, revealed decreases in *diap1*-lacZ (Fig. 2i) and *ban*-lacZ (Fig. 2j). Consistent with the previous study[39], the *ter94*[k15502] homozygous clones exhibited reduced size, possibly due to decreased Yki activity. Furthermore, overexpression of wild-type Ter94 was able to restore the reductions in *diap1*-lacZ (Fig. 2k) and fj-lacZ (Fig. 2l) induced by 35608, as well as the decreases caused by Ter94AA (Fig. 2m, n). In summary, these results demonstrate that loss of *ter94* suppresses the expression of Yki targets, and leads to growth disadvantage.

### Cytoplasmic hVCP protein enables to rescue *ter94* RNAi-induced small wings
After demonstrating the necessity of Ter94 in maintaining proper wing size, we further investigated the functional conservation of Ter94/VCP. By conducting rescue assays with a human VCP transgenic fly, we found that the small wings caused by *ter94*-RNAi (Fig. 3a–c) were restored by expressing human VCP (Fig. 3e, f). Similar to wild-type Ter94, overexpression of hVCP alone did not impact wing size (Fig. 3d). These observations suggest that hVCP is able to substitute for Ter94 in modulating wing size.

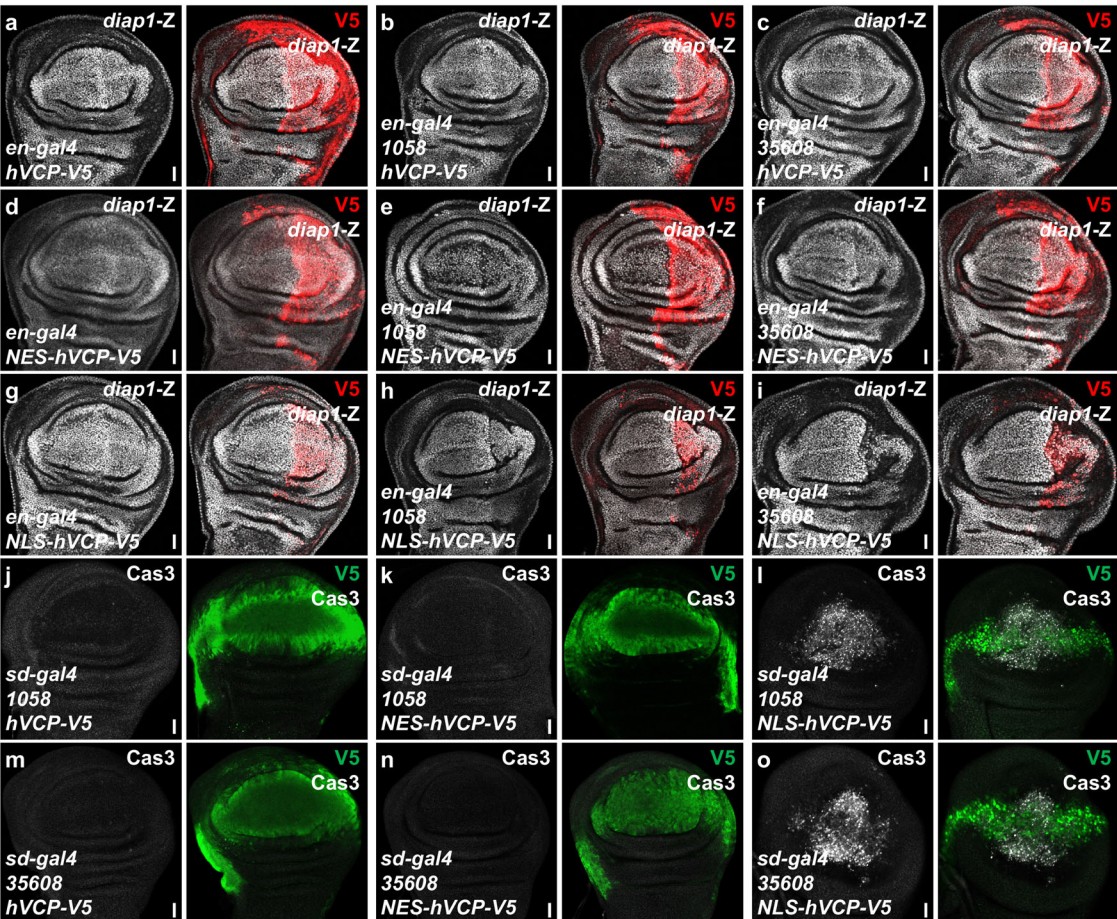

**Fig. 4 | Cytoplasmic hVCP recovers the decrease of *diap1*-lacZ by *ter94* knockdown. a–c** Wing discs expressing hVCP alone **a**, hVCP plus *ter94* RNAi **b, c** were stained to show V5 (red) and *diap1*-lacZ (white). hVCP rescues *diap1*-lacZ expression under *ter94* knockdown background. **d–f** Wing discs expressing NES-hVCP alone (**d**), NES-hVCP plus *ter94* RNAi **e, f** were stained to show V5 (red) and *diap1*-lacZ (white). **g–i** Wing discs expressing NLS-hVCP alone **g**, NLS-hVCP plus *ter94* RNAi **h, i** were stained to show V5 (red) and *diap1*-lacZ (white). **j–o** Wing discs expressing *ter94* RNAi plus hVCP **j, m**, *ter94* RNAi plus NES-hVCP **k, n**, or *ter94* RNAi plus NLS-hVCP **l, o** were stained to reveal V5 (green) and Cas3 (white). As shown, the activation of Caspases by *ter94* knockdown is rescued by NES-hVCP, not by NLS-hVCP. Scale bars: 20 μm for all wing discs.

Previous studies have shown that Ter94 is involved in the degradation of both cytoplasmic and nuclear proteins through the proteasome pathway[41]. Immunostaining revealed that both V5-Ter94 and hVCP-V5 were present in the cytoplasm and nucleus (Supplementary Figs. 3a, b). To investigate whether cytoplasmic or nuclear Ter94 regulates wing size, we constructed transgenic flies expressing NES-hVCP and NLS-hVCP, which contain a nuclear export signal (NES) and a nuclear localization signal (NLS) respectively. Immunostaining confirmed that NES-hVCP-V5 exclusively resided in the cytoplasm (Supplementary Fig. 3c), while NLS-hVCP-V5 localized in the nucleus (Supplementary Fig. 3d). Overexpression of NES-hVCP or NLS-hVCP alone did not alter wing size (Fig. 3g–i). Remarkably, NES-hVCP successfully restored the small wings caused by *ter94* knockdown (Fig. 3j, k, m, n), whereas NLS-hVCP did not (Fig. 3l, o), indicating that cytoplasmic Ter94 is important for regulating wing size.

### Cytoplasmic hVCP rescues Yki activity suppression and apoptosis caused by *ter94* knockdown

Given the above data showed that the Ter94/VCP plays a conserved role in regulating wing size, we sought to investigate whether VCP rescues *ter94* RNAi-induced suppression of Yki target genes. Although overexpression of hVCP did not affect *diap1*-lacZ (Fig. 4a), it could restore the decreased *diap1*-lacZ caused by *ter94* knockdown (Fig. 4b, c). In addition, NES-hVCP enabled to rescue *ter94* RNAi-induced *diap1*-lacZ decreases (Fig. 3d–f), whereas NLS-hVCP failed to do so (Fig. 3g–i), together suggesting that

cytoplasmic Ter94 plays a more important role in regulating the Hippo pathway.

After discovering that knockdown of *ter94* triggers apoptosis, we proceeded to test whether hVCP could inhibit this process. The results showed that hVCP was able to block *ter94* RNAi-induced apoptosis (Fig. 3j, m). Furthermore, cytoplasmic hVCP effectively inhibited the apoptosis caused by *ter94* knockdown (Fig. 3k, n), while nuclear hVCP could not (Fig. 3l, o). Overall, Ter94/VCP plays a conserved role in regulating the Hippo pathway, and cytoplasmic Ter94 is important in this regulation.

### Ter94 sits upstream of the core kinase cascade to control the Hippo pathway

Having demonstrated that loss of Ter94 decreases wing size and down-regulates Yki target gene expression, we aimed to investigate the underlying mechanism. Central to the Hippo pathway is a kinase module, through which upstream signals converge on the transcriptional cofactor Yki, leading to the coordination of target gene expression[42]. By studying genetic interactions between Ter94 and key components of the Hippo pathway, we gained insight into how Ter94 modulates wing size. Because manipulating the Hippo pathway activity throughout the wing would result in deformation, we chose *ptc*-gal4 to drive transgene expression specifically between vein L3 and vein L4[43]. Compared to the control wing (Fig. 5a), over-expression of Ter94AA using *ptc*-gal4 resulted in a noticeable decrease in the L3/L4 intervein size (Fig. 5b). Consistent with previous findings,

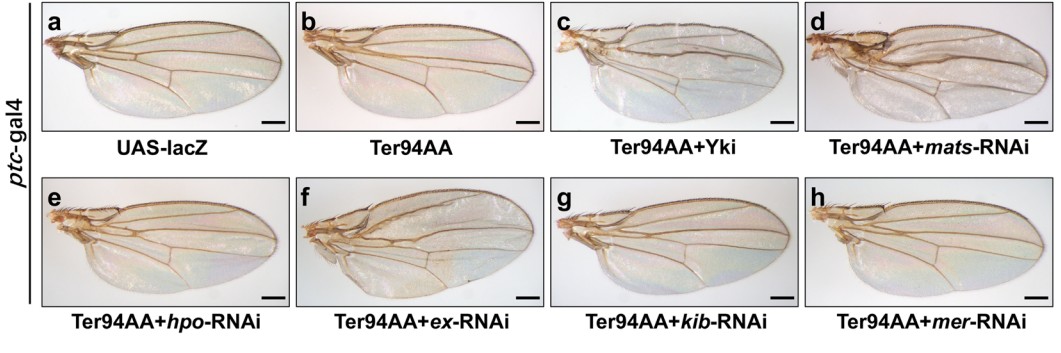

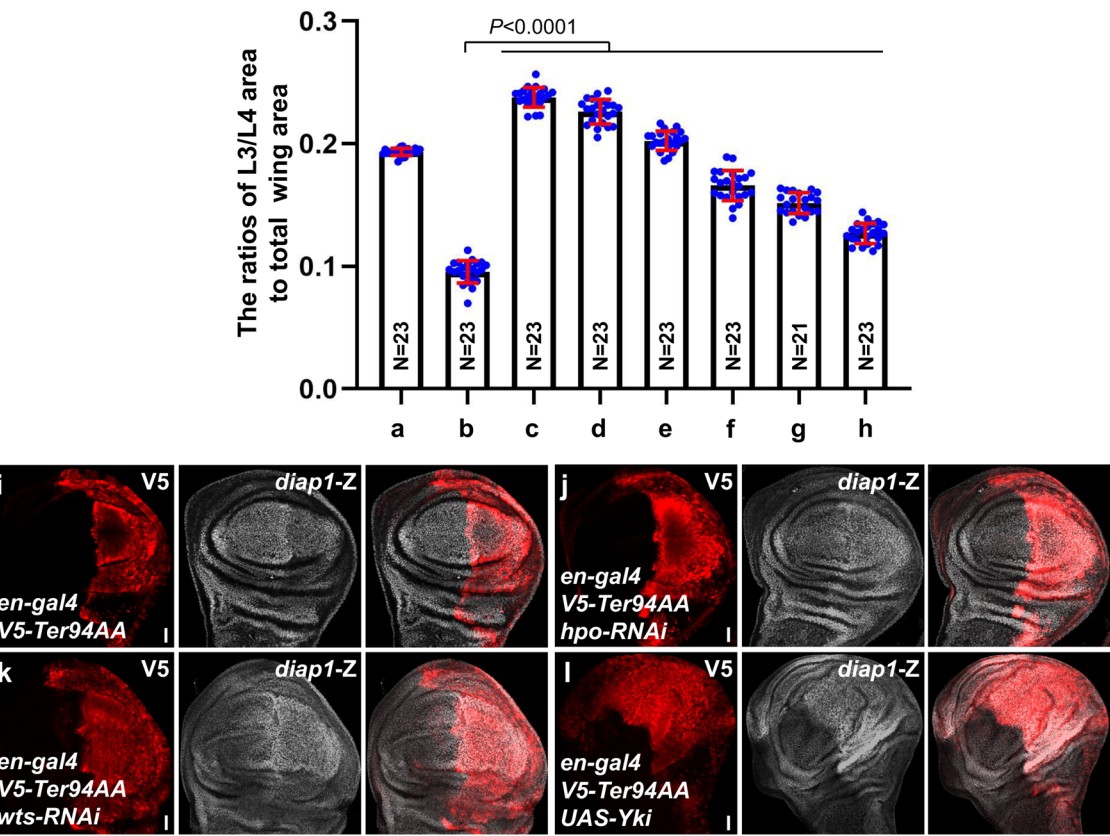

**Fig. 5 | Ter94 acts upstream of the core kinase cascade. a–h** Comparison of adult wings from control **a**, expressing Ter94AA by *ptc*-gal4 **b**, simultaneously expressing Ter94AA plus Yki **c**, expressing Ter94AA plus *mats* RNAi **d**, expressing Ter94AA plus *hpo* RNAi **e**, expressing Ter94AA plus *ex* RNAi **f**, expressing Ter94AA plus *kib* RNAi **g**, and expressing Ter94AA plus *mer* RNAi **h**. The *ptc*-gal4 drives UAS transgenes to express between vein L3 and L4. Quantification analyses were shown below. The numbers in the bars represented the number of wings counted. Scale bars: 200 μm for all adult wings. **i–k** Wing discs expressing V5-Ter94AA alone by *en*-gal4 **i**, co-expressing V5-Ter94AA plus *hpo* RNAi **j**, expressing V5-Ter94AA plus *wts* RNAi **k**, and expressing V5-Ter94AA plus Yki **l** were stained to reveal V5 (red) and *diap1*-lacZ (white). Scale bars: 20 μm for all wing discs.

overexpression of Yki or knockdown of Hippo pathway components, including *mats*, *hpo*, *ex*, *kib* and *mer*, increased the width of L3/L4 (Supplementary Fig. 4). Overexpression of Yki enabled the restoration of Ter94AA-induced undergrowth (Fig. 5c), indicating that Ter94 localizes upstream of Yki. This result further corroborated the notion that cytoplasmic Ter94 is more important for regulating the Hippo pathway. In addition, knockdown of core kinase module components, including *mats* and *hpo* also rescued the growth defect caused by Ter94AA (Fig. 5d, e), suggesting that Ter94 functions upstream of the core kinase cascade. We did not obtain a result for *wts*-RNAi since *wts* knockdown leads to larval lethality. Inhibition of the Ex-Mer-Kib branch only partially recovered the wing growth defect induced by Ter94AA (Fig. 5f–h), inferring that Ter94 may function in parallel with this complex.

After observing that Ter94 is positioned upstream of the core kinase cascade in controlling wing size, we proceeded to assess Yki activity using *diap1*-lacZ as a readout. In comparison to the Ter94AA-overexpressing wing disc (Fig. 5i), simultaneous knockdown of *hpo* or *wts* successfully rescued the decreased *diap1*-lacZ (Fig. 5j-k). Furthermore, co-expression of Yki also restored Ter94AA-induced downregulation of *diap1*-lacZ (Fig. 5l). Taken together, these epistatic analyses indicate that Ter94 functions upstream of the core kinase cassette in regulating Yki activity.

**Ter94 physically interacts with Mer**
Previous studies have demonstrated that Ter94 primarily recognizes ubiquitin-modified proteins to deliver them to the proteasome for proteolysis. Given that our above results indicated that Ter94 likely acts in parallel

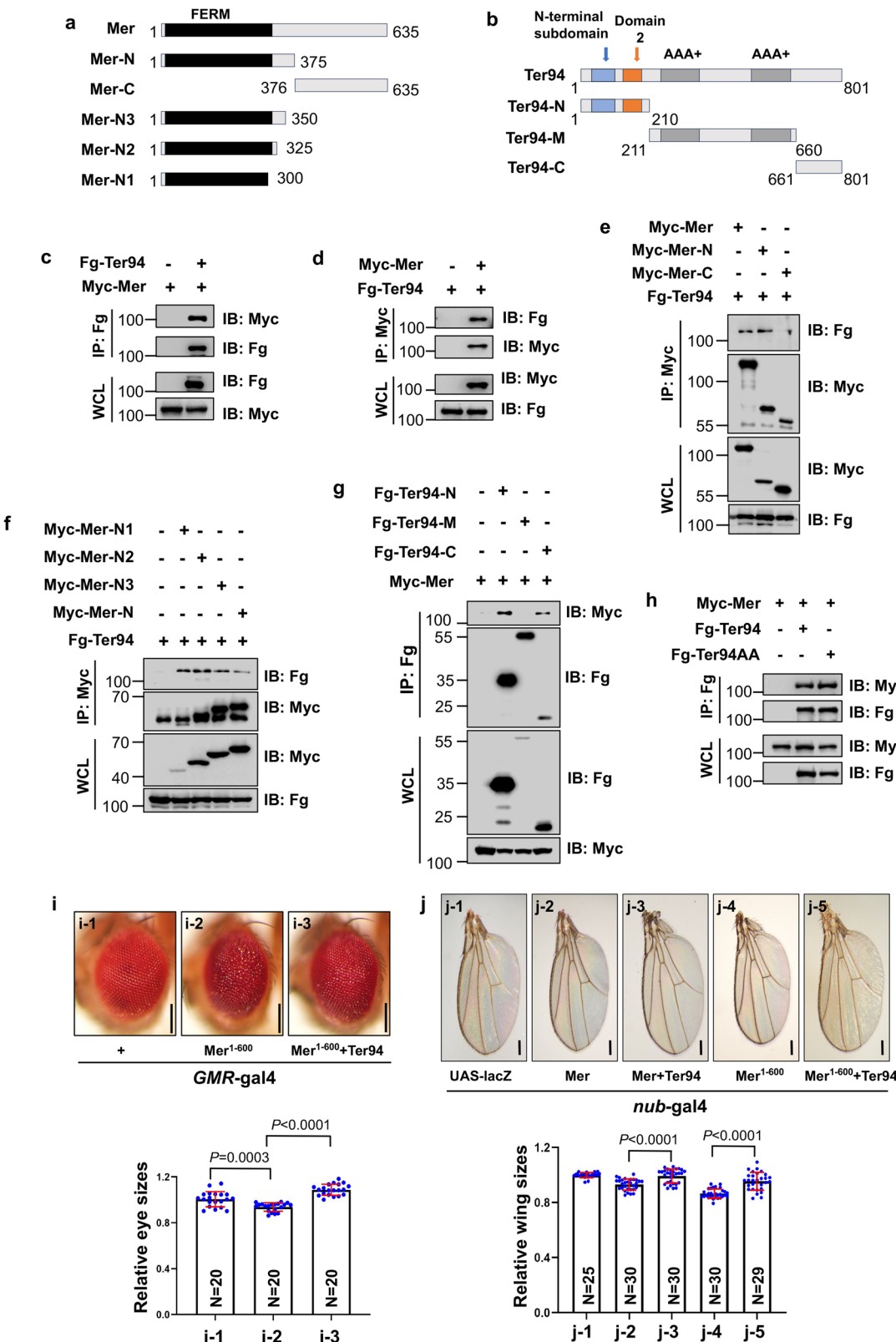

**Fig. 6 | Ter94 binds Mer to inhibit its anti-growth effect. a, b** The schematic diagrams show the domains of Mer and Ter94, and their truncated constructs used in the following co-IP. **c** Fg-Ter94 pulled down Myc-Mer in HEK-293T cells. **d** Myc-Mer pulled down Fg-Ter94 in HEK-293T cells. **e** Myc-Mer interacted with Fg-Ter94 through its N terminus. **f** The FERM domain in Mer enabled to pull down Fg-Ter94. **g** Both Fg-Ter94-N and Fg-Ter94-C were able to pull down Myc-Mer. **h** Fg-Ter94 and Fg-Ter94AA showed identical affinity to Myc-Mer. **i** Comparison of adult eyes from control (**i-1**), expressing Mer$^{1-600}$ alone via *GMR*-gal4 (**i-2**), simultaneously

expressing Mer$^{1-600}$ and Ter94 (**i-3**). Quantification analyses of relative eye sizes were shown. The numbers in the bars represented the number of eyes counted. Scale bars: 100 μm for all adult eyes. **j** Comparison of adult wings from control (**j-1**), over-expression of Mer by *nub*-gal4 (**j-2**), co-expressing Mer plus Ter94 (**j-3**), expressing Mer$^{1-600}$ alone (**j-4**), and co-expressing Mer$^{1-600}$ plus Ter94 (**j-5**). Quantification analyses of relative wing size has been shown. The numbers in the bars represented the number of wings counted. Scale bars: 200 μm for all adult wings.

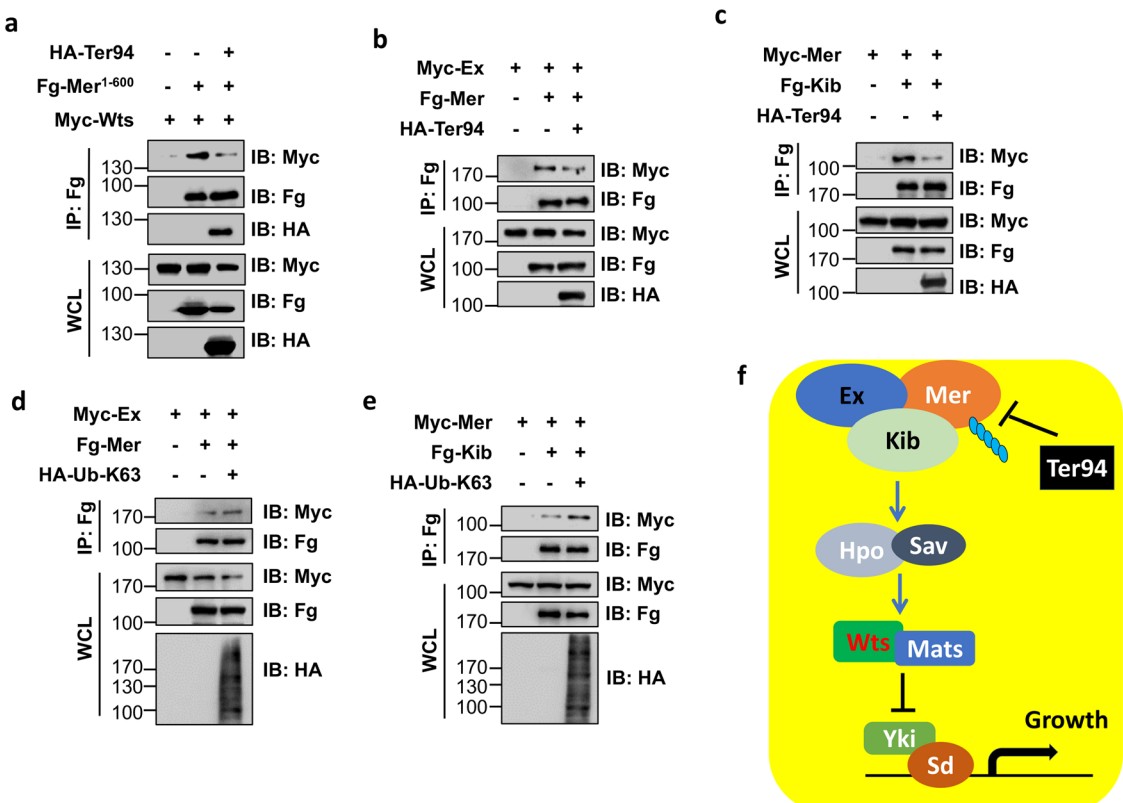

**Fig. 7 | Ter94 disrupts the Ex-Mer-Kib complex. a** HA-Ter94 decreased the interaction between Fg-Mer[1-600] and Myc-Wts. **b** HA-Ter94 suppressed Fg-Mer binding to Myc-Ex. **c** HA-Ter94 diminished the interaction between Fg-Kib and Myc-Mer. **d** Co-transfection of Ub-K63 promoted Mer binding to Ex.

**e** Co-expression of Ub-K63 enhanced the interaction between Kib and Mer. **f** A proposed model of Ter94 inhibiting the Hippo pathway. Ter94 recognized K63-linked ubiquitinated Mer to inhibit it from forming the Ex-Mer-Kib complex, thereby suppressing the initiation of the Hippo pathway.

to the Ex-Mer-Kib complex, we needed to examine the interaction between Ter94 and this complex. Three aspects point to Mer as the most likely binding partner of Ter94. First, co-immunoprecipitation (co-IP) and subsequent mass spectrometry analyses have revealed that Mer can pull down Ter94[44]. Additionally, compared to other components of the Hippo pathway, Mer exhibits dramatic ubiquitination[21], which is a prerequisite for Ter94 recognition[41]. Finally, proteomic profiling of VCP substrates in mammalian cells indicates that Mer is a candidate[34]. Thus, we examined the interaction between Ter94 and Mer through co-IP assays. As expected, Myc-Mer reciprocally bound Fg-Ter94 (Fig. 6c-d). However, Yki or Hpo did not bind to Ter94 (Supplementary Figs. 5d-e), suggesting that Ter94 specifically interacts with Mer. Since the above findings demonstrate that human VCP can replace Ter94 in regulating the Hippo pathway and wing size, we tested whether human VCP binds to Mer or its human homolog NF2. The co-IP results displayed that hVCP interacts with Mer and NF2 (Supplementary Figs. 5a-c). Mer comprises a FERM domain in its N-terminus, which is important for mediating protein-protein interactions[45]. To explore whether the FERM domain is involved in Mer-Ter94 interaction, we generated a series of truncated mutants (Fig. 6a). The co-IP results revealed that Mer binds to Ter94 via its N-terminus (Fig. 6e), with the FERM domain being sufficient for this interaction (Fig. 6f). NF2 is a well-known tumor suppressor, with high-frequency mutations in its FERM domain. Several point mutations (L46R, F62S, L64P, L141P) have been shown to abolish the anti-tumor role of NF2[46]. By sequence alignment, we strikingly found that these sites are conserved in Mer. Therefore, we mutated the corresponding sites and tested the interaction between these mutants and Ter94. As shown in Supplementary Fig. 5f, all mutants revealed weaker interactions with Ter94.

On the other hand, to map the fragment of Ter94 responsible for binding Mer, we constructed three nonoverlapping truncated mutants

(Fig. 6b). The co-IP assays showed that both the N-terminus and C-terminus of Ter94 were able to interact with Mer, while the ATPase domains failed to bind Mer (Fig. 6g). Furthermore, Ter94 and Ter94AA exhibited equivalent affinities for Mer (Fig. 6h), providing an explanation as to why Ter94AA plays a dominant-negative role.

Having demonstrated the interaction between Mer and Ter94, we next explored whether Ter94 regulates the anti-growth activity of Mer. It is known that wild-type Mer forms an auto-inhibitory structure, that can be relieved by deleting its C-terminal 35 amino acids[47]. Ectopic expression of Mer[1-600] by *GMR*-gal4 slightly decreased eye size and led to roughness, which was recovered by co-expression of Ter94 (Fig. 6i). Similarly, overexpression of Mer using *nub*-gal4 mildly decreased wing size, which was rescued by co-expressing Ter94 (Fig. 6j). In line with the previous finding[47], Mer[1-600] overexpression resulted in smaller wings, but this effect was restored by co-expression of Ter94 (Fig. 6j). These results indicate that Ter94 has the ability to suppress Mer activity.

Given that Ter94 primarily directs proteins to proteasome-mediated proteolysis[23,48], it was necessary to test whether Ter94 promotes Mer degradation. Due to the unavailability of a commercial Mer antibody, we generated a *tub*-Myc-Mer transgenic fly that expressed Myc-tagged Mer protein under the *tubulin* promoter. The Myc antibody was used to detect Myc-Mer protein levels, which were found to be evenly expressed in the wing disc (Supplementary Fig. 6a). Overexpression of *mer* RNAi was able to diminish Myc-Mer protein, confirming the reliability of the *tub*-Myc-Mer fly (Supplementary Fig. 6b). Surprisingly, knockdown of *ter94* did not impact Myc-Mer protein levels (Supplementary Figs. 6c-d), and overexpression of Ter94AA (Supplementary Fig. 6e) or wild-type Ter94 (Supplementary Fig. 6f) also had no effect. Previous studies have demonstrated the importance of the apical localization of Mer in activating the Hippo pathway[49]. Therefore, we investigated whether Ter94 influences the

localization of Mer. While Mer is typically found colocalizing with the apical domain marker Dlg (Supplementary Fig. 6g), overexpression of Ter94 resulted in a decrease in the apical positioning of Mer within epithelial cells (Supplementary Fig. 6h). In sum, Ter94 binds to the FERM domain of Mer to suppress its activity, without affecting its protein abundance.

## Ter94 dissociates the Ex-Mer-Kib complex

Previous studies have demonstrated that Mer forms a complex with Ex and Kibra at the apical domain of cells to recruit Wts for phosphorylation, ultimately activating the Hippo pathway[20,47,50]. In view of the interaction between Ter94 and Mer, we attempted to investigate whether Ter94 interferes with the formation of Mer-containing complexes. Mer[1-600] recruits Wts to the cell membrane via its N-terminal FERM domain, resulting in Wts phosphorylation and subsequent activation[47]. Interestingly, our results showed that co-expression of Ter94 decreased the interaction between Mer[1-600] and Wts (Fig. 7a). In addition, we observed that Ter94 was able to inhibit the binding of Mer to Ex (Fig. 7b) and Kib (Fig. 7c). Since the formation of an Ex-Mer-Kib complex is crucial for activating the Hippo pathway[18], our findings suggest that Ter94 may suppress the pathway by dissociating this complex.

Several previous studies have revealed that Mer undergoes non-degradative polyubiquitination, which alters its interactions with partners[21,22]. As K63-linked polyubiquitination is always involved in the regulation of protein-protein interactions, we explored whether this modification affects the assembly of the Ex-Mer-Kib complex. To eliminate the influence of proteolysis caused by ubiquitination, we opted for the Ub-K63 mutant. This mutant replaces all lysines (Ks) except K63 in ubiquitin with arginines, leaving only K63 to form K63-linked polyubiquitin chains. Remarkably, co-transfection of Ub-K63 enhanced the interaction between Mer and Ex (Fig. 7d) as well as Kib (Fig. 7e), suggesting that ubiquitinated Mer prefers to form a complex with Ex and Kib. In conclusion, our study proposes a possible mechanism in which Ter94 recognizes ubiquitinated Mer and prevents it from forming the Ex-Mer-Kib complex, leading to inactivation of the Hippo pathway (Fig. 7f).

## Discussion

Organ size determination is a complex and interesting biological process that is regulated by multiple mechanisms, with the Hippo pathway playing a key role. The Hippo pathway was initially discovered in *Drosophila* through mutagenesis screening[51]. Mutation of several components of this pathway leads to organ overgrowth[16]. Central to the pathway is the Hpo-Wts kinase module, which is activated by the upstream Ex-Mer-Kib complex[19]. Previous studies have revealed that ubiquitin modification on Mer is an important step for its activation, rather than leading to its degradation[21,22]. However, the mechanism for terminating the activity of ubiquitinated Mer has remained elusive. In this study, through genetic screening, we identified that Ter94 positively regulates wing size dependent on its ATPase activity. Knockdown of *ter94* decreased wing size, and downregulated the expression of Yki target genes. Human VCP was able to restore *ter94* RNAi-induced growth defect and Yki target gene inhibition. Furthermore, cytoplasmic Ter94 was more important for regulating wing size and the Hippo pathway. Based on epistatic analyses, we fingered out that Ter94 acts in parallel with the Ex-Mer-Kib complex to modulate the Hippo pathway. Mechanistically, Ter94 recognized the ubiquitinated Mer to prevent it from forming the Ex-Mer-Kib complex, thereby suppressing the Hippo pathway. This study reveals a mechanism to cease the activity of ubiquitinated Mer without promoting its proteolysis.

Although knockdown of *ter94* decreases wing size, and depletion of *ter94* leads to growth disadvantage, overexpression of Ter94 does not lead to an increase in wing size or activation of Yki target genes. These observations can be attributed to two reasons. First, endogenous Ter94 is sufficient to regulate Mer and maintain the normal activity of the Hippo pathway. Consistent with this possibility, overexpression of Ter94 indeed rescues the small wing and eye induced by Mer[1-600]. Alternatively, only a small portion of Mer is subject to ubiquitin modification, a key requirement for Ter94

recognition. A previous study showed that Ter94 prefers to bind K11-linked polyubiquitinated Ci[32], while another study found that Ter94/VCP recognizes K6-linked polyubiquitinated c-MYC[34]. In this study, Ter94 possibly binds to K63-linked polyubiquitin chains attached to Mer. Hence, Ter94 is able to recognize distinct polyubiquitin linkages depending on different substrates. In contrast to the canonical role of Ter94, it fails to modulate the stability of Mer. How Ter94 coordinates its degradative and non-degradative roles on different substrates will be an interesting research direction.

Mer is a renowned tumor suppressor, as its somatic mutations have been tightly linked to the development of several types of tumors, particularly schwannomas and meningiomas[52,53]. A meta-analysis has revealed that most tumor-derived Mer mutations cluster in its FERM domain[54], which is responsible for binding Ter94. It would be beneficial to investigate whether these mutations disrupt Mer binding to Ter94, thereby relieving the inhibitory effect of Ter94. As a matter of fact, inhibition of Ter94/VCP is considered to be a promising strategy for tumor intervention[55,56]. Several Ter94/VCP inhibitors, including CB-5083 and CB-5339 are under clinical trials[57]. In the further, it is necessary to explore whether Ter94/VCP inhibitors exert anti-tumor effects by activating the Hippo pathway. Thus, this study facilitates to dissect the mechanism of Ter94/VCP inhibitors inhibiting tumor progression and provides guidance for their clinical application.

## Materials and methods

### *Drosophila* genetics

*nub*-gal4, *en*-gal4, *hh*-gal4, *sd*-gal4, *ptc*-gal4, *GMR*-gal4, *ap*-gal4, *hpo*-RNAi, *wts*-RNAi, UAS-Yki, UAS-lacZ, *diap1*-lacZ, *ban*-lacZ and *fj*-lacZ have been described in our previous studies[14,58–60]. *ter94*-RNAi (1058, THU3262), *mats*-RNAi (THU3571), *hpo*-RNAi (THU0551), *ex*-RNAi (TH201501137.S), *kib*-RNAi (THU3065), *mer*-RNAi (THU2845) were purchased from TsingHua Fly Center (THFC). *ter94*-RNAi (35608), *tub*-gal80ts were obtained from Bloomington *Drosophila* Stock Center (BDSC). UAS-V5-Ter94, UAS-V5-Ter94AA, UAS-hVCP-V5, UAS-NES-hVCP-V5, UAS-NLS-hVCP-V5 transgenic flies were purchased from Core Facility of *Drosophila* Resource and Technology, Shanghai Institute of Biochemistry and Cell Biology, Chinese Academy of Sciences. UAS-HA-Mer and UAS-HA-Mer[1-600] transgenic flies were kindly from Prof. Shian Wu, Nankai University. The attB-*tub*-Myc-Mer construct was made by cloning a full-length *mer* cDNA inserted into downstream of the *α*-tubulin promoter, then inserting this construct into 25C6 attP locus (#25709, BDSC)[59].

### DNA constructs

To generate Myc-Mer, Fg-hVCP, Myc-NF2, Myc-Yki, HA-Hpo, Myc-Wts, Myc-Ex, Fg-Kib, Fg-Ter94 and HA-Ter94 constructs, we amplified the corresponding cDNA fragments using Vazyme DNA polymerase (P505), and inserted them into pcDNA3.1-Myc, pcDNA3.1-Fg or pcDNA3.1-HA backbone vectors respectively. Truncated constructs including Myc-Mer-N (aa1-375), Myc-Mer-C (aa376-635), Myc-Mer-N1 (aa1-350), Myc-Mer-N2 (aa1-325), Myc-Mer-N3 (aa1-300), Fg-Mer[1-600], Fg-Ter94-N (aa1-210), Fg-Ter94-M (aa211-660), and Fg-Ter94-C (aa661-801) were made by inserting the corresponding coding sequences into pcDNA3.1-Myc or pcDNA3.1-Fg vectors. Fg-Ter94AA, Myc-Mer-L36R, Myc-Mer-F52S, Myc-Mer-L54P, Myc-Mer-L135P and HA-Ub-K63 were made by PCR-based site-directed mutagenesis.

### Immunostaining and confocal

Immunostaining of wing discs was carried out according to our previous protocols[61]. Briefly, third-instar larvae were dissected in PBS and fixed in 4% PFA at room temperature for 20 min, then permeabilized with PBT (PBS supplemented with 0.1% Triton X-100) for three times. Larvae were incubated with primary antibodies in PBT at 4 °C for at least 4 hr, then washed with PBT for three times and incubated with fluorophore-conjugated secondary antibodies for 2 hr at room temperature. After washing for three times with PBT, discs were separated and mounted with 40% glycerol. Images were captured by Zeiss confocal microscope. Primary antibodies

used in this study included: mouse anti-V5 (1:500, MBL, M215-3), rabbit anti-cleaved Caspase-3 (1:200, Cell Signaling Technology, 9661 S), rabbit anti-β-Galactosidase (1:500, MBL, PM049), mouse anti-Myc (1:200, Santa Cruz, sc-40), rat anti-Ci (1:10, DSHB, 2A1), mouse anti-Dlg (1:10, DSHB, 4F3), rat anti-HA (1:200, Santa Cruz, sc-53516). To mark cell nuclei, wing discs were stained with DAPI (1:10000, Santa Cruz, sc-24941) for 15 min before mounting. All secondary antibodies used in this study were bought from Jackson ImmunoResearch, and were diluted at 1:500.

### BrdU labeling
Wing discs were incubated with 30 μM BrdU (Sigma, HY-15910) for 45 min in S2 medium (Hyclone) before fixation, and the subsequent immunostaining was performed according the standard protocol. Primary antibodies used in this study was mouse anti-BrdU (1:10, DSHB, G3G4).

### RNA isolation, reverse transcription, and real-time PCR
Wing discs for *ter94*-RNAi (1058, 35608, THU3262) driven by *nub*-gal4 lysed in TRIzol (Invitrogen) for RNA isolation following standard protocols. 500 ng RNA were used for reverse transcription by MonScript™ product line (Monad) according to the instructions. Real-time PCR was performed on ZY/VQ-100A (Yuanzan) using the ChamQ SYBR qPCR Master Mix (Q711, Vazyme). 2-ΔΔCt method was used for relative quantification. The primer pairs used was follows: *ter94*, 5'-AAG CTG GCC ATC CGA CAG-3' (forward), 5'-ATG GCC TCC TCG AAG TGG G-3' (reverse); *actin*, 5'-GTA CCC CAT TGA GCA CGG TA-3' (forward) and 5'-ACT CCT GCT TGC TGA TCC AC-3' (reverse). All RT-qPCR results are presented as means ± SD (standard deviation) of values from at least three experiments.

### Cell culture, transfection, and immunoblot
All cell-based assays in this study were carried out in HEK-293T cells. HEK-293T cells were maintained in Dulbecco's modified Eagle's medium (DMEM). Transfection was performed using PEI (Sigma) according to the manufacturer's instructions. 48 h after transfection, cells were collected for subsequent co-IP and IB according to our previous described[62]. The following antibodies were used for IP and IB: mouse anti-Fg (1:500 for IP, 1:5000 for IB, Sigma, F3165); mouse anti-Myc (1:200 for IP, 1:2000 for IB, Santa Cruz, sc-40); mouse anti-HA (1:2000 for IB, Santa Cruz, sc-7392); goat anti-mouse HRP (1:10000, Abmax). Uncropped blots are shown in Supplementary Fig. 7 and Supplementary Fig. 8.

### Statistics and reproducibility
Sizes of wings and eyes were measured by Image J software. Statistical analyses were performed with GraphPad Prism software, using one-way ANOVA. All data were presented as means ± SD (standard deviation), and $P < 0.05$ was considered statistically significant. Quantitative analyses were shown, with the numbers in the bars indicating the number of wings and eyes that were counted. All source data underlying the graphs are presented in Supplementary Data. All wing disc images were captured by selecting three images with consistent trends of change. The WB data shown in the article are representative and have been repeated three times.

### Reporting summary
Further information on research design is available in the Nature Portfolio Reporting Summary linked to this article.

### Data availability
The numerical source data behind the graphs can be found in the Supplementary data file. All other data are available from the corresponding author on reasonable request.

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

## Acknowledgements

We sincerely thank Dr. Dezhen Peng, the founder of Yuanzan Lifescience Co., Ltd, for donating a quantitative PCR instrument (ZY/VQ-100A) to our lab. We also appreciate Prof. Shian Wu (Nankai University, China) and Prof. Junzheng Zhang (China Agricultural University, China) for generous providing *Drosophila* stocks. We also appreciate Shanghai Institute of Biochemistry and Cell Biology, Bloomington *Drosophila* Stock Center (BDSC) and TsingHua Fly Center for providing flies, and Developmental Studies Hybridoma Bank (DSHB) at the University of Iowa for providing antibodies. This study was supported by grants from the National Natural Science Foundation of China (32270522, 32272945 and 32350710192) and the project of Double Thousand Plan in Jiangxi Province of China (09030049).

## Author contributions

The authors have made the following declarations about their contributions: Z.Z. and Q.L. designed the experiments. M.L., W.D., Y.D. and Y.Z. performed the experiments. Z.Z. and Q.L. carried out data analysis. M.L. and Z.Z. wrote the manuscript with the help of all authors.

## Competing interests

The authors declare no competing interests.
