## [Peer Review File · Communications Biology]

Reviewers' comments:

Reviewer #1 and Reviewer #3 (Remarks to the Author):

In this paper, Li et al. investigate the role of the ATPase Ter94 in wing size regulation through its interaction with Merlin, a core component of the Hippo signaling pathway. The manuscript presents intriguing genetic and physical interactions between Ter94 and Mer, providing evidence for its significance. However, several major concerns regarding proper citations, conflicting results, experimental design, and ambiguous conclusions need to be addressed. Additionally, minor issues related to terminology and incomplete sentences should be corrected.

Major #1: Proper Citations and References

The manuscript lacks proper citations for the key mutant form (Ter94AA) of Ter94. The ATP-binding sites (K248 and K521) were replaced by alanine and have been extensively utilized by researchers (PLoS Genet. 7(2): e1001288., PLoS Genet. 10(9): e1004675., Nat. Commun. 12(1): 4258., Development 150(14): dev201557.) And another version (an ATPase-deficient VCP transgene VCP QQ) was also utilized in multiple papers (Development (2011) 138 (6): 1153–1160., Nature 414, 652-656). Including references for their extensive use is crucial to avoid misleading readers and to clarify the novelty of these ideas. The nature of Ter94AA, whether it acts as a dominant negative allele or not, should be clarified, and relevant references should be provided.

Major #2: Conflicting Results and Diap1 lacZ

The Diap1 lacZ results conflict with a previous publication (Development (2011) 138 (6): 1153–1160), necessitating additional experiments like western blotting or Diap1 staining. Specifically, in the previous paper, the authors showed that the endogenous DIAP1 protein level decreases with the expression of wild-type VCP (Ter94) but the level was increased with the expression of dominant-negative form VCP QQ (Fig6A of the paper) in the S2 cell. They also showed physical interaction between DIAP1 and Ter94 (Fig.6C of the paper).

The authors are encouraged to explore the genetic interaction between Diap1 and Ter94 or discuss potential relationships in the Discussion section.

Major #3: Figure 5 Experiments and Choice of Gal4 Line

Clarification is needed regarding why Ter94AA lines were used without Ter94 RNAi lines in Figure 5. The choice of ptc-Gal4 for this experiment requires additional justification. The authors are encouraged to include rescue experiments using Ter94 RNAi lines and other Gal4 lines like nub-Gal4 or en-Gal4 for comprehensive analysis.

Major #4: Clarification of Ter94 Binding to Ubiquitinated Mer

The manuscript lacks clarity on why the authors concluded that Ter94 binds to ubiquitinated Mer. A proper rationale is needed, and biochemical assays comparing Ter94–Mer binding affinity with and without ubiquitination could strengthen the argument.

Major #5: Terminology Regarding Mer Complex Interaction

The claim that Ter94 'extracts' Mer from the Mer-Ex-Kib complex lacks sufficient experimental evidence.

The authors may perform additional experiments to prove that. Alternatively, using the term 'inhibit' might be more appropriate.

Minor #1: Clarification of ATPase Activity Importance

In line 78, it is suggested that the ATPase activity of Ter94 is dispensable. This should be clarified to convey that the ATPase activity is essential or crucial for Ter94 function based on the evidence provided or other references.

Minor #2: Explanation of Screening in Line 103

In line 103, the authors should explain the type of screening performed and the purpose behind it for better contextual understanding.

Minor #3: Clarification of Line 111

In line 111, it is unclear why the authors concluded that endogenous Ter94 is sufficient to maintain proper wing sizes based on Fig 1D and 1G. Additional clarification or referencing is needed.

Minor #4: Completion of Sentence in Line 246

In line 246, the sentence is incomplete. Please provide the necessary completion for clarity.

Reviewer #2 (Remarks to the Author):

In this manuscript, Li et al. demonstrated that the ATPase Ter94 regulates wing sizes by disrupting the Ex-Mer-Kib complex, which in turn inhibits the canonical Hippo pathway. Knockdown of Ter94 triggered apoptosis and reduced the expression of Yki target genes. The authors employed RNAi and mutant clones to investigate the role of Ter94 in Hippo pathway regulation. The formation of the Ex-Mer-Kib complex is a crucial step in activating Hippo signaling, but the mechanisms controlling complex formation remain unknown. Additionally, the authors showed that human VCP can substitute for Ter94 in regulating wing size, suggesting functional conservation during evolution. Overall, this manuscript is well-written and presents mostly solid results with significant novelty, making it a strong candidate for publication in *Communications Biology*. However, there are several issues that need to be addressed to further enhance the quality of the manuscript.

1. Although this study primarily focuses on wing size regulation, it is worth investigating whether Ter94 also modulates the size of other organs, such as the eye, particularly in the context of Hippo activation.
2. In Fig 1, the knockdown of Ter94 or the overexpression of Ter94AA in wing discs induces apoptosis. However, it is important to determine the effects of these manipulations on cell proliferation, as reduced cell proliferation could contribute to smaller wing sizes.

3. Previous studies have highlighted the significance of apical localization of Mer in activating the downstream Hippo pathway. The authors should test whether Ter94 affects the apical localization of Mer.

4. To rule out the potential off-target effects, the authors should examine the knockdown efficiencies of the distinct RNAi lines used in this study.

5. In Fig 3 and 4, the authors demonstrate that human VCP can functionally substitute Ter94 in regulating Hippo signaling and wing size. It would be interesting to further investigate whether human VCP binds to Ter94 or its human homolog NF2, thus elucidating the functional conservation.

6. Ter94 binds to the N-terminal FERM domain of Mer to suppress Ex-Mer-Kib complex formation. In humans, the homolog of Mer, NF2, is a well-known tumor suppressor, with many mutations clustered in its FERM domain in human tumor samples. It would be intriguing to explore if these mutations affect the affinity of NF2 with Ter94.

Reviewers' comments:

Reviewer #1 and Reviewer #3 (Remarks to the Author):

In this paper, Li et al. investigate the role of the ATPase Ter94 in wing size regulation through its interaction with Merlin, a core component of the Hippo signaling pathway. The manuscript presents intriguing genetic and physical interactions between Ter94 and Mer, providing evidence for its significance. However, several major concerns regarding proper citations, conflicting results, experimental design, and ambiguous conclusions need to be addressed. Additionally, minor issues related to terminology and incomplete sentences should be corrected.

Response: Thanks a lot for your nice evaluation on our manuscript. We have carried out additional experiments to address all your concerns. Also, we have modified the manuscript according to your suggestions. We sincerely hope this revised manuscript will be satisfactory to you.

Major #1: Proper Citations and References

The manuscript lacks proper citations for the key mutant form (Ter94AA) of Ter94. The ATP-binding sites (K248 and K521) were replaced by alanine and have been extensively utilized by researchers (PLoS Genet. 7(2): e1001288., PLoS Genet. 10(9): e1004675., Nat. Commun. 12(1): 4258., Development 150(14): dev201557.) And another version (an ATPase-deficient VCP transgene VCP QQ) was also utilized in multiple papers (Development (2011) 138 (6): 1153–1160., Nature 414, 652-656). Including references for their extensive use is crucial to avoid misleading readers and to clarify the novelty of these ideas. The nature of Ter94AA, whether it acts as a dominant negative allele or not, should be clarified, and relevant references should be provided.

Response: Thanks for your suggestions. We have read the related papers carefully, and

added new references in our revised manuscript.

Major #2: Conflicting Results and Diap1 lacZ

The Diap1 lacZ results conflict with a previous publication (Development (2011) 138 (6): 1153–1160), necessitating additional experiments like western blotting or Diap1 staining. Specifically, in the previous paper, the authors showed that the endogenous DIAP1 protein level decreases with the expression of wild-type VCP (Ter94) but the level was increased with the expression of dominant-negative form VCP QQ (Fig6A of the paper) in the S2 cell. They also showed physical interaction between DIAP1 and Ter94 (Fig.6C of the paper).

The authors are encouraged to explore the genetic interaction between Diap1 and Ter94 or discuss potential relationships in the Discussion section.

Response: Thanks very much for your constructive suggestions. In our previous paper, we generated a *tub*-Myc-Diap1 transgenic fly, in which Myc-tagged Diap1 was expressed by *tubulin* promoter (1). Surprisingly, knockdown of *ter94* using 1058 or 35608 failed to change Myc-Diap1 level (A-B, below). In addition, overexpression of wild-type Ter94 or Ter94AA did not affect Myc-Diap1 (C-D, below). Paradoxically, a previous paper has demonstrated that Ter94 was able to promote Diap1 proteolysis in S2 cells (2). One possible explanation is that Ter94-mediated Diap1 degradation is tissue-specific or cell-specific.

Major #3: Figure 5 Experiments and Choice of Gal4 Line

Clarification is needed regarding why Ter94AA lines were used without Ter94 RNAi lines in Figure 5. The choice of ptc-Gal4 for this experiment requires additional justification. The authors are encouraged to include rescue experiments using Ter94 RNAi lines and other Gal4 lines like nub-Gal4 or en-Gal4 for comprehensive analysis.

Response: Thanks a lot for your advice. As a matter of fact, we have performed these genetic experiments using other Gal4 drivers. As shown below, the small wing caused by *ter94* knockdown using *nub-gal4* was rescued by co-expressing *hpo*-RNAi, *wts*-RNAi or *ex*-RNAi. Possible due to the wide expression pattern of *nub-gal4*, knockdown of some components of the Hippo pathway always leads to wing deformation, even adult lethality. Hence, we chose *ptc-gal4*, only expresses between vein L3 and L4 of the wing, for these genetic analyses. In addition, we utilized *diap1-lacZ* staining to validate

our results. Overexpression of Ter94AA by *en-gal4* reduced *diap1-lacZ* (Figure 5I), which was rescued by *hpo*-RNAi (Figure 5J), *wts*-RNAi (Figure 5K) or UAS-Yki (Figure 5L), suggesting that Ter94 sits upstream of the core kinase cascade to modulate Yki activity.

Major #4: Clarification of Ter94 Binding to Ubiquitinated Mer

The manuscript lacks clarity on why the authors concluded that Ter94 binds to ubiquitinated Mer. A proper rationale is needed, and biochemical assays comparing Ter94–Mer binding affinity with and without ubiquitination could strengthen the argument.

Response: Thanks a lot. Many previous studies have clearly demonstrated that Ter94 recognizes ubiquitin-modified proteins and extracts them for subsequent processing (3). We carried out additional experiments and found that the interaction of Ter94 and Mer was robustly enhanced by HA-Ub-K63 (below).

Major #5: Terminology Regarding Mer Complex Interaction

The claim that Ter94 ‘extracts’ Mer from the Mer-Ex-Kib complex lacks sufficient experimental evidence. The authors may perform additional experiments to prove that. Alternatively, using the term ‘inhibit’ might be more appropriate.

Response: Thank you for your advice. Indeed, the term “inhibit” is more appropriate. According to your suggestion, we have modified the description in this revised manuscript.

Minor #1: Clarification of ATPase Activity Importance

In line 78, it is suggested that the ATPase activity of Ter94 is dispensable. This should be clarified to convey that the ATPase activity is essential or crucial for Ter94 function based on the evidence provided or other references.

Response: We sincerely appreciate the valuable comments. We have checked the literature carefully and added more references (Nat Cell Biol. 13(11):1376-82., Front Mol Biosci.4,39.) into the Introduction part in the revised manuscript.

Minor #2: Explanation of Screening in Line 103

In line 103, the authors should explain the type of screening performed and the purpose

behind it for better contextual understanding.

Response: We have added our screening purpose in the revised manuscript that we want to discover some genes that affect organ size.

Minor #3: Clarification of Line 111

In line 111, it is unclear why the authors concluded that endogenous Ter94 is sufficient to maintain proper wing sizes based on Fig 1D and 1G. Additional clarification or referencing is needed.

Response: We overexpressed wild type Ter94 in the wing using distinct Gal4 drivers, but did not lead to overgrowth. Based on these observations, we think that endogenous Ter94 has reached the growth demand.

Minor #4: Completion of Sentence in Line 246

In line 246, the sentence is incomplete. Please provide the necessary completion for clarity.

Response: We apologize for our mistake. We have corrected it in this revised manuscript.

Reviewer #2 (Remarks to the Author):

In this manuscript, Li et al. demonstrated that the ATPase Ter94 regulates wing sizes by disrupting the Ex-Mer-Kib complex, which in turn inhibits the canonical Hippo pathway. Knockdown of Ter94 triggered apoptosis and reduced the expression of Yki target genes. The authors employed RNAi and mutant clones to investigate the role of Ter94 in Hippo pathway regulation. The formation of the Ex-Mer-Kib complex is a crucial step in activating Hippo signaling, but the mechanisms controlling complex formation remain unknown. Additionally, the authors showed that human VCP can

substitute for *Ter94* in regulating wing size, suggesting functional conservation during evolution. Overall, this manuscript is well-written and presents mostly solid results with significant novelty, making it a strong candidate for publication in *Communications Biology*. However, there are several issues that need to be addressed to further enhance the quality of the manuscript.

Response: Thanks a lot for your nice evaluation on our manuscript.

1. Although this study primarily focuses on wing size regulation, it is worth investigating whether *Ter94* also modulates the size of other organs, such as the eye, particularly in the context of Hippo activation.

Response: Thank you for your suggestions. Compared with the control eye, overexpression of *Mer-aal-600* resulted in a small eye, which was restored by co-expressing *Ter94* (Figure 6I). We carried out additional experiments to validate the role of *Ter94* in eye size determination. As shown below, V5-Wts-induced small eyes were rescued by *Ter94*. Taken together, *Ter94* is able to modulate eye sizes in Hippo activation context.

2. In Fig 1, the knockdown of *Ter94* or the overexpression of *Ter94AA* in wing discs induces apoptosis. However, it is important to determine the effects of these manipulations on cell proliferation, as reduced cell proliferation could contribute to

smaller wing sizes.

Response: Thanks for your suggestions. In this revised manuscript, we examined the proliferation using BrdU incorporation assay. Compared with the control wing (Figure S1E), neither knockdown of *ter94* (Figures S1F-G) nor overexpression of *Ter94AA* (Figure S1H) decreased BrdU signals, indicating the proliferation is not decreased. Strikingly, *Ter94AA* overexpression upregulated BrdU incorporation (Figure S1H), possibly due to apoptosis-induced compensatory proliferation.

3. Previous studies have highlighted the significance of apical localization of Mer in activating the downstream Hippo pathway. The authors should test whether Ter94 affects the apical localization of Mer.

Response: This is a good question. Previous studies have clearly demonstrated that the apical localization of Mer is important for it activating the Hippo pathway. In this revised manuscript, we performed additional experiments to check the localization of Mer protein. Cross-section (X-Z) views of HA-Mer staining showed that overexpression of *Ter94* indeed reduced the apical localization of Mer (Figures S6G-H).

4. To rule out the potential off-target effects, the authors should examine the knockdown efficiencies of the distinct RNAi lines used in this study.

Response: Thank you for your advice. To remove the possible off-target effects, we added an additional RNAi (THU3262), and got the same results (Figures S2A-D). Compared with the control wing disc (Figure S2A), knockdown of *ter94* using THU3262 apparently activated apoptosis (Figure S2B), which was completely rescued by co-expressing *Ter94* (Figure S2C). Furthermore, THU3262 also downregulated *fj-lacZ* expression (Figure S2D). In sum, we utilized three non-overlapping RNAi to knock down *ter94*, and got the same phenotype. These observations were enough to

rule out off-target effects. According to your suggestion, we extracted RNA from wing discs expressing indicated RNAi by *nub-gal4*. The RT-qPCR results showed that three RNAi lines were able to silence endogenous *ter94* (Figure S2E).

5. In Fig 3 and 4, the authors demonstrate that human VCP can functionally substitute Ter94 in regulating Hippo signaling and wing size. It would be interesting to further investigate whether human VCP binds to Ter94 or its human homolog NF2, thus elucidating the functional conservation.

Response: Thank you for your suggestions. Co-IP assay results showed that hVCP was able to bind Mer or NF2 (Figures S5A-C).

6. Ter94 binds to the N-terminal FERM domain of Mer to suppress Ex-Mer-Kib complex formation. In humans, the homolog of Mer, NF2, is a well-known tumor suppressor, with many mutations clustered in its FERM domain in human tumor samples. It would be intriguing to explore if these mutations affect the affinity of NF2 with Ter94.

Response: Thanks! NF2 is a well-known tumor suppressor, with high-frequency mutations on its FERM domain. Several point mutations (L46R, F62S, L64P, L141P) abolished the anti-tumor role of NF2 (4). By sequence alignment, we strikingly found these sites are conserved in Mer. Therefore, we mutated the corresponding sites and tested the interaction between mutants and Ter94. As shown in Figure S5F, all mutants revealed weaker interaction with Ter94. In this study, we found that Ter94 was a negative regulator of Mer. These mutations possibly reduce Mer binding to Ter94, relieving Ter94's inhibitory role.

References:

1. Liu B, Ding Y, Sun B, Liu Q, Zhou Z, Zhan M. The Hh pathway promotes cell apoptosis through Ci-Rdx-Diap1 axis. *Cell Death Discov.* 2021;7(1):263.
2. Rumpf S, Lee SB, Jan LY, Jan YN. Neuronal remodeling and apoptosis require VCP-dependent degradation of the apoptosis inhibitor DIAP1. *Development.* 2011;138(6):1153-60.

3. Ye Y. Diverse functions with a common regulator: ubiquitin takes command of an AAA ATPase. *J Struct Biol.* 2006;156(1):29-40.
4. Li W, You L, Cooper J, Schiavon G, Pepe-Caprio A, Zhou L, et al. Merlin/NF2 suppresses tumorigenesis by inhibiting the E3 ubiquitin ligase CRL4(DCAF1) in the nucleus. *Cell.* 2010;140(4):477-90.

Reviewers' comments:

Reviewer #1 (Remarks to the Author):

The authors did not address the issue raised during the previous review regarding Major #1: "... The nature of Ter94AA, whether it acts as a dominant negative allele or not, should be clarified, and relevant references should be provided."

In line 118, the authors explained that the reduced wing size suggests the dominant-negative role of Ter94AA. However, the possibility that Ter94AA functions as a gain-of-function allele cannot be ruled out with the current data. Figures 1I and 1P, which show that overexpression of wildtype Ter94 partially rescues the wing size reduction/cell death phenotype, could support the idea only when a proper control experiment is performed. The observed phenotypic rescue could simply be caused by the dilution of Gal4, not by Ter94 overexpression. I suggest the authors perform additional control experiments such as nub>Ter94AA with a neutral control (lacZ or mCherry) and sd>Ter94AA with a neutral control (lacZ or mCherry).

All other concerns have been addressed satisfactorily.

Reviewer #2 (Remarks to the Author):

The authors have adequately addressed all of my concerns regarding this manuscript. Therefore, I wholeheartedly support the publication of this work in Communications Biology.

Reviewer #3 (Remarks to the Author):

The authors did not address the issue raised during the previous review regarding Major #1: "... The nature of Ter94AA, whether it acts as a dominant negative allele or not, should be clarified, and relevant references should be provided."

In line 118, the authors explained that the reduced wing size suggests the dominant-negative role of Ter94AA. However, the possibility that Ter94AA functions as a gain-of-function allele cannot be ruled out with the current data. Figures 1I and 1P, which show that overexpression of wildtype Ter94 partially rescues the wing size reduction/cell death phenotype, could support the idea only when a proper control experiment is performed. The observed phenotypic rescue could simply be caused by the dilution of Gal4, not by Ter94 overexpression. I suggest the authors perform additional control experiments such as nub>Ter94AA with a neutral control (lacZ or mCherry) and sd>Ter94AA with a neutral control (lacZ or mCherry).

All other concerns have been addressed satisfactorily.

Reviewers' comments:

Reviewer #1 and Reviewer #3 (Remarks to the Author):

The authors did not address the issue raised during the previous review regarding Major #1: "... The nature of Ter94AA, whether it acts as a dominant negative allele or not, should be clarified, and relevant references should be provided."

*In line 118, the authors explained that the reduced wing size suggests the dominant-negative role of Ter94AA. However, the possibility that Ter94AA functions as a gain-of-function allele cannot be ruled out with the current data. Figures 1I and 1P, which show that overexpression of wildtype Ter94 partially rescues the wing size reduction/cell death phenotype, could support the idea only when a proper control experiment is performed. The observed phenotypic rescue could simply be caused by the dilution of Gal4, not by Ter94 overexpression. I suggest the authors perform additional control experiments such as *nub*>Ter94AA with a neutral control (*lacZ* or *mCherry*) and *sd*>Ter94AA with a neutral control (*lacZ* or *mCherry*).*

All other concerns have been addressed satisfactorily.

Response: Thanks very much for your suggestions. We have added related references in our revised manuscript (*Dev Cell.* 25(6): 636-44., *Biol Open.* 8(1): bio038984.). It is clearly stated in these references that Ter94AA acts as a domain-negative mutant.

In order to avoid that the observed phenotype is due to dilution of Gal4, we followed your advice and performed additional experiments. Compared with the control (A, below), overexpression of Ter94AA driven by *nub-gal4* increased the Cas3 level (B, below). Overexpression of Ter94AA driven by *nub-gal4* with a neutral control (GFP) displayed a same result (C, below). Co-expression of the wild-type Ter94 to some extent rescued Ter94AA-induced upregulated Cas3 level (D, below).

nub> Ter94AA or *nub*> Ter94AA with a neutral control (GFP) significantly reduced

the wing size, co-expression of Ter94 partially rescues the small wings caused by Ter94AA (E-H, below).

REVIEWERS' COMMENTS:

Reviewer #1 (Remarks to the Author):

We have no other comments. Those are satisfactorily addressed.

Reviewer #3 (Remarks to the Author):

All concerns were addressed. I would be happy to support the publication.